# Comparison of DNA Extraction and Amplification Techniques for Use with Engorged Hard-Bodied Ticks

**DOI:** 10.3390/microorganisms10061254

**Published:** 2022-06-20

**Authors:** Gretchen C. Reifenberger, Bryce A. Thomas, DeLacy V. L. Rhodes

**Affiliations:** Department of Biology, Berry College, Mount Berry, GA 30149, USA; grettie.reifenberger@vikings.berry.edu (G.C.R.); bryce.thomas@vikings.berry.edu (B.A.T.)

**Keywords:** ticks, DNA extraction, PCR, tick-borne diseases, engorged ticks

## Abstract

Tick-borne infections are a serious threat to humans, livestock, and companion animals in many parts of the world, often leading to high morbidity and mortality rates, along with decreased production values and/or costly treatments. The prevalence of the microbes responsible for these infections is typically assessed by the molecular identification of pathogens within the tick vectors. Ticks sampled from animals are often engorged with animal blood, presenting difficulties in the amplification of nucleic acids due to the inhibitory effects of mammalian blood on the enzymes used in polymerase chain reactions (PCRs). This study tested two tick preparation methods, three methods of DNA extraction, and four commercially available DNA polymerases to determine the most reliable method of extracting and amplifying DNA from engorged ticks. Our study found that the phenol–chloroform extraction method yielded the highest concentration of DNA, yet DNA extracted by this method was amplified the least successfully. Thermo Scientific’s Phusion Plus PCR Master Mix was the best at amplifying the tick 16s rRNA gene, regardless of extraction method. Finally, our study identified that using the Qiagen DNeasy Blood & Tissues kit for DNA extraction coupled with either Phusion Plus PCR Master Mix or GoTaq DNA polymerase Master Mix is the best combination for the optimized amplification of DNA extracted from engorged ticks.

## 1. Introduction

Tick-borne diseases (TBD) comprise serious and prevalent vector-borne infections in many parts of the world and affect human, livestock, and companion animal populations [1,2,3]. To better understand the prevalence of tick-borne pathogens and the risks associated with contracting one, studies often rely on pathogen detection within tick vectors. To this end, large samples of ticks are collected through a variety of methods including flagging or dragging common tick habitats, setting up baited CO_2_ traps, or from living/dead animal hosts. Different methods of tick collection have been shown to be biased towards collecting certain species and/or life stages, underscoring the necessity of using a variety of collection methods to optimize sampling efforts [4,5,6]. Once ticks have been collected, total DNA is extracted from each tick, and molecular techniques such as polymerase chain reaction (PCR) and reverse line blot hybridization are used to identify the pathogens being carried by the tick [7,8,9,10].

When ticks are collected from animals, they are often attached to their host and in the process of taking a bloodmeal (also known as feeding). Female ticks can become engorged with the blood of their host during this process, leading to swollen bodies and an increase in size [11]. When total DNA is extracted from these engorged ticks, DNA from the tick, any pathogens they harbor, and the host(s) they have fed on is collected. Previous studies have shown that components of mammalian blood, specifically immunoglobulin G and hemoglobin, can be inhibitory to PCR [12,13,14]. Though several recent studies have used both engorged and unengorged ticks, many studies have chosen to focus on unengorged/flat ticks when evaluating the prevalence of tick-borne pathogens, despite the aforementioned sampling biases [8,10,15,16]. To gain a more comprehensive view of the prevalence of tick-borne pathogens, developing a methodology for extracting and amplifying DNA from engorged ticks is necessary.

Few studies have experimentally evaluated different protocols for studying pathogen prevalence in engorged ticks. In this study, we used engorged ticks collected from hunter-captured deer to compare common DNA extraction methodologies and DNA polymerases with the purpose of identifying the most reliable combination of protocols for working with engorged ticks. Additionally, we compared two different methods of processing ticks prior to DNA extraction. For DNA extraction methods, we chose to evaluate a commercially available spin column kit, phenol–chloroform extraction, and cetyltrimethylammonium bromide extraction, assessing the DNA concentration and purity of each method. For the amplification of DNA, we used primers directed against the tick 16s rRNA gene to assess the ability of four different DNA polymerases to amplify DNA extracted by these three methods. In this paper, we present findings comparing these preparation and extraction methods and DNA polymerases to provide guidance on how to best utilize engorged ticks in studies assessing the prevalence of tick-borne pathogens.

## 2. Materials and Methods

### 2.1. Tick Collection

Ticks of all life stages were collected from hunted deer during the fall 2018 hunting season (September–December) on Fort Campbell Army Base in Fort Campbell, Kentucky, USA. Hunting on the army base is managed by the Fort Campbell Fishery and Wildlife Service, and all harvested deer are checked in and recorded by this office. All deer were combed for ticks while being checked in. Collected ticks were placed in vials containing 95% ethanol, one vial per deer.

### 2.2. Tick Preparation

All ticks selected for this study were adult female ticks at various levels of engorgement. Tick species was not determined due to the difficulty of using morphological characteristics to identify engorged ticks [17,18]. The weight and length of each engorged tick was recorded prior to preparation. Ticks were prepared for extraction by two methods: either whole ticks were quartered and the whole tick was used for extraction, or the abdominal contents were removed and used for extraction alone. For our first preparation method, whole ticks were cut into quarters, horizontally and medially, and placed in sterile 1.5 mL microcentrifuge tubes [19]. For our second method, all abdominal contents were obtained by cutting open the tick and scraping the internal organs and bloodmeal contents out of the tick with a scalpel and into a sterilized 1.5 mL microcentrifuge tube. Between ticks, scalpels and tweezers were cleaned with bleach to prevent cross-contamination.

### 2.3. DNA Extraction

Three different extraction methods were compared in this study: extraction using phenol–chloroform [20], a commercially available Qiagen DNeasy Blood & Tissues Kit [21], and cetyltrimethylammonium bromide extraction [22]. For all extractions, the quantity and purity of samples were determined by spectrophotometry (Thermo Scientific NanoDrop 2000 Spectrophotometer, Thermo Fisher Scientific, Waltham, MA, USA). The samples were stored at 4 °C until further use. The protocols for each extraction method are detailed below and supplied as handouts in Appendix A.

#### 2.3.1. Phenol–Chloroform

Phenol–chloroform extraction was chosen for this study because it has been shown to support the amplification of DNA using tick 16s rRNA primers [23]. Phenol–chloroform DNA extraction was carried out as previously described [24]. Following tick preparation, 200 µL of proteinase K (0.2 mg/mL; Sigma-Aldrich, St. Louis, MI, USA) was added to each tick and samples were incubated for 16 h at 55 °C. Samples were vortexed to disrupt any remaining solids and centrifuged at 14,000 RPM for 10 min. The supernatant was transferred to a clean tube, and 400 µL of phenol–chloroform–isoamyl alcohol mixture (Sigma-Aldrich, St. Louis, MI, USA) was added. The samples were vortexed to mix and centrifuged at 14,000 RPM for 2 min, and the upper aqueous phase was transferred to a clean tube. This step was performed twice. Following the second centrifugation and passage of the aqueous phase to a clean tube, 400 µL of 3 M sodium acetate (pH 5.2; VWR International, Solon, OH, USA) and 1200 µL of cold 100% ethanol were added and mixed by inversion. Extracted DNA was precipitated at −20 °C for at least 2 h or at −70 °C for at least 30 min. The DNA was pelleted by centrifugation at 14,000 RPM for 10 min. The ethanol was then removed, and the pellet was washed with 500 µL of 70% ethanol. The samples were centrifuged again, the ethanol removed, and the DNA pellet was allowed to air dry before being resuspended in 50 µL of water for molecular biology (Millipore, Billerica, MA, USA).

#### 2.3.2. Qiagen DNeasy Blood & Tissues Kit

The commercially available DNeasy Blood & Tissues kit (Qiagen, Germantown, ML, USA) was utilized in this study according to the manufacturer’s protocol, following the instructions for purification of total DNA from animal tissues. Following tick preparation, 180 µL of Buffer ATL and 20 µL of proteinase K were added to the samples, which were then incubated at 56 °C overnight. The next day, the samples were spun down at 14,000 RPM for 10 min and the supernatant was transferred to a new tube. Next, 200 µL of Buffer AL was added and the samples were vortexed thoroughly, followed by the addition of 200 µL ethanol (96–100%) and another round of vortex. The samples were then pipetted into a kit spin column and centrifuged at 8000 RPM for 1 min. The flow through was discarded and 500 µL of Buffer AW1 was added, removed by centrifugation for 1 min at 8000 RPM, and discarded. This step was repeated using the same volume of Buffer AW2, followed by centrifugation for 3 min at 14,000 RPM. The spin column was then transferred to a clean tube, and 200 µL of Buffer AE was added to the center of the spin column membrane and incubated for 1 min at room temperature to allow for the elution of the extracted DNA. DNA was then collected by centrifugation for 1 min at 8000 RPM.

#### 2.3.3. Cetyltrimethylammonium Bromide (CTAB)

The CTAB protocol utilized was adapted from the Ops Diagnostics CTAB Method with Spin Columns [25]. For CTAB extraction, 500 µL of CTAB Extraction Buffer (Ops Diagnostics, Lebanon, NJ, USA) was added to prepared tick samples, which were then placed in a 60 °C water bath for 30 min. After incubation, samples were centrifuged at 10,000 RPM for 5 min to pellet any residual tick components, and the supernatant was transferred to a clean tube. Next, five µL of RNase A (10 mg/mL) was added to the supernatant and the mixture was vortexed for 5 s and then incubated at room temperature for 15 min. Following incubation, 350 µL of 100% isopropanol (2-propanol) was added to each sample, which were then vortexed for 5 s and incubated at −20 °C for 15 min. After incubation, the mixture was transferred to a spin column (Qiagen, Germantown, ML, USA) and centrifuged at 8000 RPM for 4 min. The flow through was discarded and 200 µL of ice cold 70% ethanol was added to the spin column before centrifuging again at 8000 RPM for 2 min. This step was repeated once before the spin column was centrifuged for 5 min at 14,000 RPM to remove any residual ethanol. The spin columns were then transferred to a clean microcentrifuge tube, and 100 µL of TE buffer was added to elute the DNA. The samples were incubated at room temperature for 5 min and then centrifuged at 8000 RPM for 1 min.

### 2.4. Polymerase Chain Reaction (PCR)

Four different DNA Polymerase reagents were tested as part of this study: GoTaq Green Master Mix (Promega, Madison, WI, USA), Phusion Plus PCR Master Mix (Thermo Scientific, Waltham, MA, USA), AmpliTaq Gold Master Mix (Applied Biosystems, Waltham, MA, USA), and Hemo KlenTaq (New England BioLabs, Ipswitch, MA, USA). The gene chosen for amplification was the 16s rRNA gene and the primers used have been published previously [26]. Following PCR, the amplified DNA was run on a 1% agarose gel, stained with ethidium bromide, and imaged using a Bio-Rad Gel Doc EZ Imager (Bio-Rad, Hercules, CA, USA). Gel images are shown in Appendix A. All reagents were used following protocols described by their respective manufacturers and run in a Bio-Rad T100 Thermal Cycler (Bio-Rad, Hercules, CA, USA). DNA extracted through the phenol–chloroform method yielded higher concentrations than the other two extraction methods; therefore, 0.5 µL of DNA from each sample was used for PCR. The exact protocol details and thermocycler conditions for each reagent are as follows:

#### 2.4.1. GoTaq Green Master Mix

Each reaction contained 12.5 µL of GoTaq Green Master Mix, 1 µL of 10 µM forward and reverse primers (0.5 µL of each), 9 µL of dH20, and 2 µL of tick DNA. Thermal cycler conditions: 94 °C for 2 min, 34 cycles of 94 °C for 30 s, 48 °C for 30 s, and 72 °C for 1 min, and a final extension at 72 °C for 4 min.

#### 2.4.2. Phusion Plus PCR Master Mix

Each reaction contained 10.5 µL of 2× Phusion Plus PCR Master Mix, 2.50 µL of 10 µM forward and reverse primers (1.25 µL of each), 8 µL of dH_2_O, and 2 µL of DNA. Thermal cycler conditions: 98 °C for 30 s, 35 cycles of 98 °C for 10 s, 60 °C 30 s, and 72 °C for 30 s, and a final extension at 72 °C for 10 min.

#### 2.4.3. AmpliTaq Gold Master Mix

Each reaction contained 12.5 µL of AmpliTaq Gold Taq Master Mix, 2 µL of 10 µM forward and reverse primers (1 µL of each), 9 µL of dH_2_O, and 2 µL of DNA. Thermal cycler conditions: 95 °C for 10 min, 35 cycles of 95 °C for 15 s, 55 °C for 30 s, and 1 min at 72 °C, and a final extension at 72 °C for 5 min.

#### 2.4.4. Hemo KlenTaq

Each reaction contained 5 µL of 5× Hemo KlenTaq reaction buffer, 0.5 µL of 10 mM dNTPs (Promega, Madison, WI, USA), 1.5 µL of 10 µM forward and reverse primers (0.75 µL of each), 2 µL of Hemo KlenTaq, 13.5 µL of dH_2_O, and 2 µL of DNA. Thermal cycler conditions: 95 °C for 3 min, 40 cycles of 20 s at 95 °C, 30 s at 60 °C, and 2 min at 68 °C, with a final extension of 68 °C for 10 min.

### 2.5. Statistical Analysis

All statistics on the DNA extraction results were calculated using R statistical software, version 4.1.3. Linear models were created for all DNA concentration data and 260/280 data, respectively, followed by an analysis of the variants of each. Post hoc Tukey tests were performed for pairwise comparisons of the ANOVA, and the significance was determined using a *p*-value ≤ 0.05 [27]. The amplification data were analyzed using Student’s *T* test in Microsoft Excel.

## 3. Results

### 3.1. DNA Concentration

Whole bodied ticks yielded higher DNA concentrations than ticks with their exoskeleton removed, regardless of the DNA extraction method used. The phenol–chloroform extraction method extracted significantly higher amounts of DNA than either the Qiagen kit (*p*_0.05_ = 0.0000012) or the CTAB method (*p*_0.05_ = 0.0000015), with an average of 2864.1 ng/µL from whole-bodied ticks and 1927.1 ng/µL from abdominal contents alone. CTAB was able to extract higher DNA concentrations than the Qiagen kit, but these results were not significant in comparison with the phenol–chloroform concentrations (Figure 1). The DNA concentration and 260/280 ratios of each sample, along with tick weight, can be found in Appendix A.

### 3.2. DNA Amplification

All four of the commercially available DNA polymerases used in this study were able to amplify DNA extracted using each extraction method tested with varying degrees of success (Table 1). Phusion Plus PCR Master Mix had the highest percentage of amplification, amplifying 84.8% of all samples, while Hemo KlenTaq had the lowest percentage with 57.6% of samples amplified, though it worked significantly better with DNA extracted through Qiagen (*p* = 0.024) and CTAB (*p* = 0.036) than through phenol–chloroform. There was no significant difference in amplification when the whole body of the tick was used in extraction versus the abdominal contents alone. AmpliTaq Gold Master Mix amplified samples extracted using the Qiagen kit or the CTAB method better than DNA extracted by phenol–chloroform, while GoTaq Green Master Mix worked well only with samples extracted with the Qiagen kit. DNA extracted using the phenol–chloroform method was able to be amplified well using Phusion Plus Master Mix, but all other DNA polymerases showed less amplification with these samples (Table 1). Across all DNA polymerases, the Qiagen kit extracted DNA that had the best overall amplification.

## 4. Discussion

Investigating the distribution of tick vectors and the prevalence of tick-borne pathogens is imperative to understanding the risk of contracting a tick-borne infection. When sampling ticks for eco-epidemiological studies, it is necessary to use multiple methods of sampling to get a fully representative cohort of ticks. Most studies investigating DNA extraction from ticks have focused on unengorged nymph or adult ticks; very few studies have focused on engorged ticks. The well-documented difficulties of amplifying DNA from engorged ticks hinder the full understanding of tick-borne pathogen prevalence. Because of this, there is a need for a clear methodology for the extraction and amplification of DNA from engorged ticks. Here, we have tested two tick preparation methods, three extraction methods, and four commercially available DNA polymerases to determine the optimal methodology for the molecular analysis of engorged ticks.

Prior to extraction, we prepared our ticks in two different ways, either by quartering whole ticks or bisecting the ticks and removing the abdominal contents for DNA extraction. Because the chitinous exoskeleton of ticks is mechanically tough and can be difficult to disrupt [8], we wanted to determine if taking the abdominal contents alone would be sufficient for DNA extraction and amplification. As most tick-borne pathogens reside in the salivary glands or midgut of the tick, the abdominal contents alone are necessary for pathogen detection. In comparing DNA extraction from whole-bodied ticks or abdominal contents alone, we found that whole ticks yield higher quantities of DNA (Figure 1). In comparing PCR amplification, there was no statistical difference in amplification between our two tick preparation methods. While preparing the ticks prior to extraction, we found removing the abdominal contents to be a difficult and laborious step. Additionally, as no efforts were made to disrupt the exoskeleton outside of quartering the ticks, our study has shown that extensive disruption of the exoskeleton through methods such as bead-beating or enzymatic lysis is not necessary. For these reasons, and because there seems to be no clear advantage to using only the abdominal contents, our recommendation is to quarter and utilize whole engorged ticks when extracting DNA.

In this study, we utilized phenol–chloroform extraction, the Qiagen DNeasy Blood & Tissues kit, and CTAB extraction to see which method was the best at extracting amplifiable DNA from engorged ticks. Each method has been documented in the literature for use with unengorged ticks [9,10,28], though their use with engorged ticks has not been investigated thoroughly. When extracting DNA from engorged ticks, both the total concentration of DNA and the purity of that DNA are important for subsequent PCR amplification [29,30]. As mammalian proteins commonly found in the blood have been shown to be inhibitory to PCR, and engorged ticks are full of mammalian blood, the ability of an extraction method to effectively separate DNA from these mammalian proteins is imperative [12,13,14]. In this study, the 260/280 ratio of each sample was determined and used to assess DNA purity. CTAB and phenol–chloroform samples produced ratios primarily within the expected range of 1.6–2.0, indicating little protein or reagent contamination. Qiagen samples, however, deviated from this range, most likely due to low concentrations of DNA (Appendix A). In all cases, 100% of the samples extracted through each method were able to be amplified, indicating the successful removal of inhibitory proteins by all three DNA extraction methods.

When assessing DNA concentration, all three methods of extraction were able to successfully extract DNA. The phenol–chloroform method produced the highest concentration of DNA by far (Figure 1). While both the Qiagen kit and CTAB method extracted on average DNA concentrations of 52.9 and 180.8 ng/µL, respectively, the phenol–chloroform method extracted concentrations above 2000 ng/µL, though there was more between-sample variability with this extraction method. When used for PCR amplification, 100% of the samples were able to be amplified by one or more DNA polymerases tested, indicating that all three DNA extraction methods can be used to extract sufficient concentrations of amplifiable DNA from engorged ticks.

To determine the best commercially available DNA polymerase for use with DNA extracted from engorged ticks, four different enzymes were tested: GoTaq Green Master Mix, Phusion Plus PCR Master Mix, AmpliTaq Gold Master Mix, and Hemo KlenTaq. Phusion Plus PCR Master Mix was the most reliable DNA polymerase, amplifying 84.8% of all tick samples and showing the least variability across DNA extraction methods (Table 1). This reagent is a high-fidelity DNA polymerase that is marketed as having fast extension times and a high tolerance for inhibitors [31]. These qualities make it an attractive option for use in studies involving engorged ticks. Conversely, Hemo KlenTaq showed the lowest amplification overall at 57.6%. This was a surprising result, as this DNA polymerase is marketed specifically for use when amplifying products from blood, without the need for DNA extraction [32]. Our other two DNA polymerases, GoTaq Green Master Mix and AmpliTaq Gold Master Mix, both fell in the middle, amplifying 66.7% and 69.7% of the samples, respectively. GoTaq Green Master Mix worked markedly better with samples extracted using the Qiagen kit, providing us with the highest amplification percentage of any of our trials (91% of all samples extracted with the Qiagen kit). With all results compared, Phusion Plus PCR Master Mix, GoTaq Green Master Mix, and AmpliTaq Gold Master Mix are all suitable DNA polymerases for the amplification of DNA extracted from engorged ticks.

This study is preliminary in nature and therefore has limitations. Our greatest limitation is our small sample size. As we were using wild-caught ticks from a single hunting season, our sample sizes per extraction method were limited. Additionally, as these were wild-caught ticks, we have no knowledge of their pathogen burden, hindering our ability to test them for specific tick-borne pathogen DNA. In the future, this study could be strengthened by the addition of more engorged ticks, specifically those that have fed on animals experimentally infected with a tick-borne pathogen.

## 5. Conclusions

In conclusion, our study has shown that it is possible to extract and amplify DNA from engorged ticks. Each DNA extraction method yielded DNA that was amplifiable by PCR, indicating that all three methods are able to sufficiently remove inhibitory mammalian blood proteins, but the Qiagen DNeasy Blood & Tissues kit and the CTAB method produced the most reliably amplifiable DNA. Additionally, there is no clear advantage to extracting DNA from the abdominal contents alone and excluding the chitinous exoskeleton. Our study has also shown that Phusion Plus Master Mix was the most reliable DNA polymerase for amplifying DNA extracted from engorged ticks, but GoTaq Green Master Mix and AmpliTaq Gold Master Mix are also suitable enzymes.

## Figures and Tables

**Figure 1 microorganisms-10-01254-f001:**
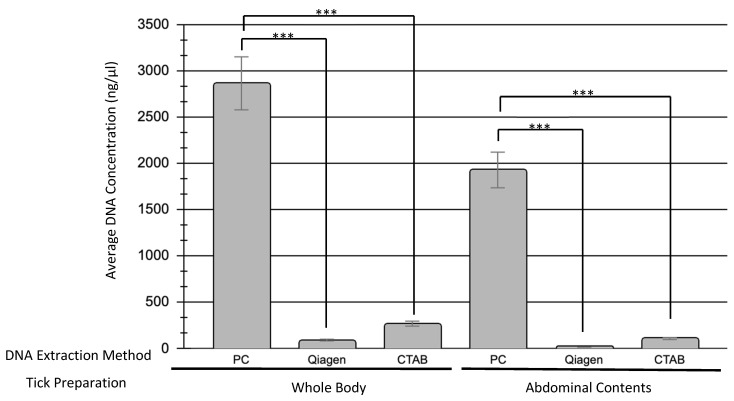
Comparison of DNA concentrations obtained by different DNA extraction methods. Three different DNA extraction methods were investigated: Phenol–chloroform (PC), Qiagen DNeasy Blood & Tissues Kit (Qiagen), and Cetyltrimethylammonium bromide (CTAB). A total of 22 samples were tested with each method, 11 of which used the whole body of the tick while the other 11 used only the abdominal contents. The average DNA concentration obtained using each method is shown here and separated by tick preparation method (whole body versus abdominal contents only). Statistical significance (*p* ≤ 0.05) is denoted by asterisks.

**Table 1 microorganisms-10-01254-t001:** Percentage amplification of DNA extracted from engorged ticks by four commercially available DNA polymerase reagents.

Extraction Method	DNA Polymerase			
Phenol–Chloroform	GoTaq	Phusion	AmpliTaq	Hemo Klen
Whole Ticks	54.5% (6/11)	81.8% (9/11)	63.6% (7/11)	36.4% (4/11)
Abdominal Contents	45.5% (5/11)	91.0% (10/11)	9.1% (1/11)	0.0% (0/11)
Total	50.0% (11/22)	86.4% (19/22)	36.4% (8/22)	18.2% (4/22)
**Qiagen**				
Whole Ticks	81.8% (9/11)	91.0% (10/11)	81.8% (9/11)	91.0% (10/11)
Abdominal Contents	100% (11/11)	81.8% (9/11)	91.0% (10/11)	63.4% (7/11)
Total	91.0% (20/22)	86.4% (19/22)	86.4% (19/22)	77.3% (17/22)
**CTAB**				
Whole Ticks	45.5% (5/11)	91.0% (10/11)	91.0% (10/11)	72.7% (8/11)
Abdominal Contents	72.7% (8/11)	72.7% (8/11)	81.8% (9/11)	81.8% (9/11)
Total	59.1% (13/22)	81.8% (18/22)	86.4% (19/22)	77.3% (17/22)
**Totals**	**66.7% (44/66)**	**84.8% (56/66)**	**69.7% (46/66)**	**57.6% (38/66)**

## Data Availability

Not applicable.

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
