# Peer review of "Comparison of DNA Extraction and Amplification Techniques for Use with Engorged Hard-Bodied Ticks"

_microorganisms, 2022, doi:10.3390/microorganisms10061254_

Round 1
Reviewer 1 Report
1. In order to extract DNA from ticks and detect pathogens, it is a prerequisite that PCR, which detects the DNA of the tick itself, is 100% successful. However, in this study, the success rate was low, and even under the conditions considered to have the highest positive rate, it is unlikely that this method can be used for investigative research. The PCR bands in the supplement figure are also unstable, with many smears and thin bands.
2. The PCR bands were cut out and the concentration was measured. However, what I would like to know is whether there was a difference in the concentration or purity of the DNA extracted from the ticks.
3. The reason why some samples were successfully PCRed while others were not is because of the variation in the tick species, isn't it?
Did you not identify the tick type by morphology? Is it possible that the primers did not match?
Author Response
Dear Reviewer #1,
Below you will find my responses to your individual comments regarding our manuscript. I have also attached a cover letter. Thank you for your time in reviewing our manuscript.
Reviewer #1: Comments and Suggestions for the Authors
Comment: In order to extract DNA from ticks and detect pathogens, it is a prerequisite that PCR, which detects the DNA of the tick itself, is 100% successful. However, in this study, the success rate was low, and even under the conditions considered to have the highest positive rate, it is unlikely that this method can be used for investigative research. The PCR bands in the supplement figure are also unstable, with many smears and thin bands.
Response: The authors thank the reviewer for their comment. We agree – when detecting pathogens in ticks, it is ideal that the PCR for tick DNA is 100% successful. This has long been the issue with using engorged ticks in pathogen prevalence research – due to the presence of inhibitors in the mammalian blood within the engorged ticks, PCR is often far less than 100% successful. The answer to this problem has been to primarily use unengorged ticks in any pathogen prevalence analysis. But, due to the biases introduced by different tick sampling methods, it is important to use engorged ticks in these analyses. This is why we chose to compare different methods of DNA extraction and different DNA polymerases specifically with engorged ticks – to find the best combination of reagents that will allow us to remove as many inhibitors as possible from engorged tick DNA, increasing the percentage of positive PCR samples. The fact that none of our methods produced 100% positive samples was not surprising. Additionally, the fact that our gels showed smearing and/or light bands was not surprising either – engorged ticks produce DNA with a fair degree of protein contamination due to all the mammalian blood inside the ticks and no DNA extraction method is able to completely remove every bit of protein contamination. Our study has produced recommendations as to DNA extraction methods and polymerases that work best with engorged ticks.
Comment: The PCR bands were cut out and the concentration was measured. However, what I would like to know is whether there was a difference in the concentration or purity of the DNA extracted from the ticks.
Response: The first sentence of the above comment is incorrect. DNA concentration and purity was measured by spectrophotometry after DNA extraction, not following PCR. PCR bands were not cut out and used to determine DNA concentration. Figure 1 shows the DNA concentrations extracted through each method and both concentration and purity of extracted DNA are included in Supplemental Table S1 and addressed in the discussion in lines 273 - 297.
Comment: The reason why some samples were successfully PCRed while others were not is because of the variation in the tick species, isn't it? Did you not identify the tick type by morphology? Is it possible that the primers did not match?
Response: We did not identify the ticks based on morphology because of the lack of distinct morphological characteristics following engorgement. A statement pertaining to this has been added to the methods section, along with references addressing the difficulties in identifying engorged ticks by morphology (Lines 74-75).
The primers used in these analyses were designed in our lab and have been used in many projects as a positive control for DNA extraction when surveying for pathogens. Additionally, they have been included in a published manuscript (see reference number 26). The primers were created by aligning the 16s rRNA gene of all the tick species that are found in the southeastern United States, where all our sampling takes place. Conserved segments of sequence were selected as these primers and the primer set was tested with DNA from all species of ticks commonly found in this region. We confidently feel that any lack of amplification was due to DNA extraction techniques not effectively removing inhibitors and/or DNA polymerases being inhibited rather than primers not finding their target.

Reviewer 2 Report
The manuscript submitted for review was written very accurately. The topic is relevant and very useful. In fact, isolating DNA from engorged ticks is challenging and often leads to failure due to contaminations. Articles that compare different methods are extremely helpful to other researchers. The selection of methods is also good, as in addition to commercially available DNA isolation kits, the authors also use the phenol-chloroform method, which although labor-intensive is considered one of the "gold" extraction methods.
Major remarks:
1Comment 1. Authors are requested to add in the results section the statistically significant differences between the different extraction kits. Also, these values can be presented in Figure 1. Line 219 also need to present the statistical difference
2. Comment 2. Line 261, this statement needs a citation. "As mammalian proteins commonly found in the blood have been shown to be inhibitory to PCR, and engorged ticks are full of mammalian blood, the ability of an extraction method to effectively separate DNA from these mammalian proteins is imperative."
1Comment 3 You don't mention what species of ticks you collected, nor the sex and stage of development. Are your results only from drunk females? Please briefly insert in the section results data on number, species, sex and stage of development.
1Comment 4. In the Discussion section, line 266 you present the protein contamination range. Add the DNA purity results of the samples. It is interesting to compare these results, I also consider them important for this manuscript.
Minor remarks I have commented in the Pdf version
· Note the spelling of the degrees. Insert a space between the numerical value and the unit symbol.
Author Response
Dear Reviewer #2,
Below you will find my responses to your individual comments regarding our manuscript. I have also attached a cover letter. Thank you for your time in reviewing our manuscript.
Reviewer #2: Comments and Suggestions for Authors
Comment: Authors are requested to add in the results section the statistically significant differences between the different extraction kits. Also, these values can be presented in Figure 1. Line 219 also need to present the statistical difference
Response: The authors have added the p values for our statistically significant results to the results sections for the DNA extraction data. These numbers can be found in Line 197.
For the statement in Line 219 (now Line 230) the word “significantly” was removed from the AmpliTaq Gold results and statistical significance data was added to the Hemo KlenTaq results (Lines 227-228).
Comment: Line 261, this statement needs a citation. "As mammalian proteins commonly found in the blood have been shown to be inhibitory to PCR, and engorged ticks are full of mammalian blood, the ability of an extraction method to effectively separate DNA from these mammalian proteins is imperative."
Response: Thank you for this comment. References have been added for this statement.
Comment: You don't mention what species of ticks you collected, nor the sex and stage of development. Are your results only from drunk females? Please briefly insert in the section results data on number, species, sex and stage of development.
Response: The authors thank the reviewers for the chance to clarify this information. All ticks selected for this study were adult female ticks that were at some stage of engorgement. This information has been added to the Tick Preparation section of the Methods and Materials (Lines 73-74). No nymphal or larval life stages were used in our analyses. The numbers of ticks used in each DNA extraction group are represented in Table 1. For each DNA extraction method, 22 ticks were used for a total of 66 ticks.
Species of each tick was not determined due to the difficulty in using morphological characteristics to identify engorged ticks. This information and references addressing these difficulties has also been added to the Methods and Materials (Lines 74-75). Ideally tick species would have been determined using molecular analyses but this study was carried out at a small liberal arts college and funding was not available for this additional work.
Comment: In the Discussion section, line 266 you present the protein contamination range. Add the DNA purity results of the samples. It is interesting to compare these results, I also consider them important for this manuscript.
Response: A table containing the DNA concentration and 260/280 ratio of each sample has been added as Supplemental Table S1 and referenced in the Results section (Lines 200-202).
Comment: Note the spelling of the degrees. Insert a space between the numerical value and the unit symbol.
Response: Thank you for bringing this error to our attention. It has now been fixed.

Round 2
Reviewer 1 Report
Comment: Are you optimizing for each enzyme in the PCR? I get the impression that the temperatures for each step of the reaction cycle and the primer sequences have not been thoroughly studied. The type of enzyme should be optimized for the crude sample, not just the one used in this study.
Response: Thank you so much for giving us the opportunity to clarify the purpose of the study detailed in our manuscript “Comparison of DNA Extraction and Amplification Techniques for use with Engorged Hard-Bodied Ticks.” Specifically in this letter, we are responding to the comment left by Reviewer 1:
We are not optimizing the reactions described in this study, only providing a comparison of existing methods of extraction and PCR amplification of DNA from engorged hard-bodied ticks, as the title of our manuscript states. As stated in lines 154-155, all PCR protocols, including temperatures used, were those provided by the manufacturer of each DNA polymerase. For primer annealing temperatures, temperatures were determined by using the online Tm calculators provided by each manufacturer. The primer sequences used in this study have been used extensively in our lab for the last 7 years, have been included in a publication, and have been shown to amplify tick DNA from all
tick species that are found in our geographic region. Please refer to our previous response to reviewer questions concerning our primer sequences to read further how these primers were developed.
We agree that a study that optimizes each enzyme would be useful, but that was not the objective of our study. We chose to provide a comparison of common methods that are used frequently with flat ticks, for the purpose of providing guidance to researchers who would like to also include engorged ticks in their analysis.
Again, thank you for giving us the opportunity to further discuss our work. If there are any more concerns with our manuscript, please do not hesitate to reach out.